# Biologically driven DOC release from peatlands during recovery from acidification

Hojeong Kang[1], Min Jung Kwon[1,4], Sunghyun Kim[1,5], Seunghoon Lee[1,6], Timothy G. Jones[2], Anna C. Johncock[2], Akira Haraguchi[3] & Chris Freeman[2]

Peatlands store 1/3 of global soil carbon, destabilisation of which contributes much to the recent increase in DOC (dissolved organic carbon) in freshwater ecosystems. One suggested mechanism for the enhanced decomposition of peat and the releases of DOC is recovery from acidification. However, no biological role in the process has yet been identified. Here we report extracellular enzyme activities and microbial composition in peatlands of Korea, the UK, Japan and Indonesia, and find higher pH to promote phenol oxidase activities, greater abundances in *Actinobacteria* and fungi, and enhanced pore-water DOC concentrations. Our pH manipulation experiments also showed that increase in pH enhanced phenol oxidase activity and DOC production with greater Actinobacterial and fungal abundances. Finally, knockout or addition of phenol oxidase dramatically changed DOC and phenolic production, indicating the central role of phenol oxidase in DOC mobilisation. Our findings provide evidence to support a previously unrecognized biological mechanism through which pH increases activate phenol oxidase, accelerating the release of DOC and phenolics.

[1] School of Civil and Environmental Engineering, Yonsei University, Seoul 03722, Korea. [2] School of Natural Sciences, Bangor University, Bangor LL57 2UW, UK. [3] The University of Kitakyushu, Kitakyushu 802-8577, Japan. [4] Present address: Korea Polar Research Institute, Inchon 21990, Korea. [5] Present address: Smithsonian Environmental Research Center, Edgewater 21037-0028 MD, USA. [6] Present address: Shine Biopharm Inc., Seoul 06536, Korea. These authors contributed equally: Hojeong Kang, Chris Freeman. Correspondence and requests for materials should be addressed to H.K. (email: hj_kang@yonsei.ac.kr) or to C.F. (email: c.freeman@bangor.ac.uk)

ncreases in dissolved organic carbon (DOC) concentrations in streams have been recorded at locations across the world including Europe, North America, and East Asia over the past few decades[1–3]. This has drawn considerable attention because DOC mobility profoundly affects the quality of drinking water, bio-availability of heavy metals and toxic chemicals, and also secondary production in aquatic ecosystems[4]. Furthermore, this may represent a transfer of upland carbon stock to river eco-systems and eventually ocean sediments at a global scale[5].

DOC is a part of dissolved organic matter (DOM) which also includes other materials such as dissolved organic nitrogen (DON) and phosphorus (DOP). However, DOC and DOM are often interchangeably used and measured by the same method. A large proportion of DOC originates from watersheds with organic soils such as peatlands, which store over a third of global soil organic carbon[6]. Proposed mechanisms for greater production and releases of DOC from peatland include more frequent droughts followed by re-flooding[7], elevated atmospheric $CO_2$ concentration[8], higher air and soil temperature[1,9], and recovery from acidification[10]. Although different drivers may function at different spatio-temporal scales[11,12], the acidification-recovery hypothesis was based not only on extensive field observations[10] but also a manipulation experiment[2]. The latter studies suggested that purely abiotic chemical releases of humic materials through changes in their solubility caused by pH changes are the main mechanism driving the greater release of DOC under higher pH conditions[13]. However, those studies have neglected potential biological linkages between pH and DOC releases, despite evidence in other fields that microbial processes are capable of driving such changes in addition to chemical reactions and physical leaching from the peat matrix. Microorganisms in soils are directly involved in decomposition of peat by producing extracellular enzymes[14]. In particular, phenol oxidase has been noted as a key enzyme playing a central role in decomposition of the peat matrix[9]. Additionally, any pH-dependent change in microbial community structure could offer a missing link in the information needed to fully appreciate the effects of pH on decomposition in peatlands where specific functionally-defined groups of microorganisms drive decomposition of the recalcitrant carbon compounds that dominate peat[15]. Various studies have, for example, shown that pH is one of the main factors that determine microbial community structures in soils[16], which potentially affects processes including decomposition.

The present study aimed to investigate biogeochemical linkages between pH increase and DOC production in peatlands. We hypothesized that rising pH would change microbial abundance and stimulate key enzyme activities involved in decomposition of peat, resulting in higher release of DOC and phenolic compounds into waters. To address this, we conducted four different studies. First, we conducted a field survey at seven peatlands representing a pH-gradient in Korea, UK, Japan and Indonesia. The sites are either located over 500 m a.s.l. or hydrologically isolated except for precipitation, allowing us to compare effects of acid deposition under natural conditions. Second, we studied a field pH manipulation experiment where peatlands were treated with acid or alkali to simulate acidification (or recovery) under natural conditions. Third, we conducted a pH manipulation experiment at the laboratory scale in which temperature and pH were manipulated and their effects on DOC and microbial properties were closely monitored over the period of a month. Finally, a short-term enzyme manipulation was performed by adding or inhibiting phenol oxidase, and changes in production of DOC and phenolic materials were observed to confirm our hypothesis. Our results indicate that pH increases promote enhanced phenol oxidase activity, greater abundances in *Actinobacteria* and fungi, and enhanced pore-water DOC concentrations.

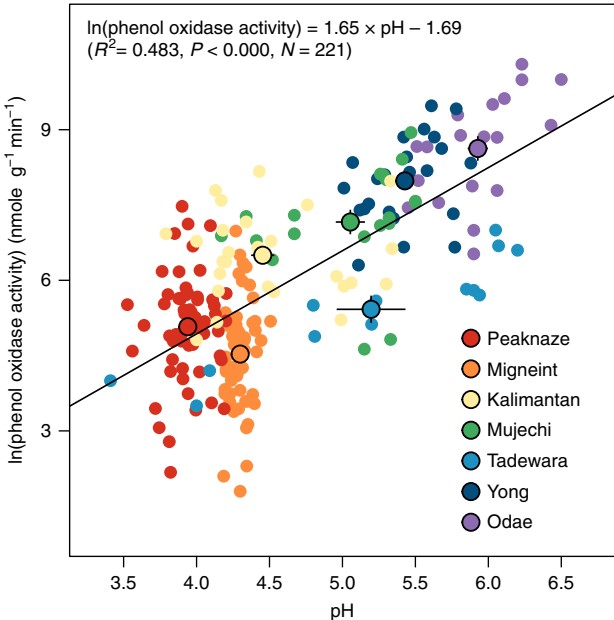

**Fig. 1** A positive correlation between pH and logarithm of phenol oxidase activities in seven peatlands. Small points are individual measurements and large points are mean values of each peatland. Error bars indicate the standard error of means

## Results and Discussion

**pH effects on phenol oxidase and DOC in global peatlands**. A positive correlation between pH and phenol oxidase activity was found across the sites. When all data were plotted logarithmically in a single graph, activity of phenol oxidase appeared to be positively associated with pH (Fig. 1; $R^2 = 0.483$, $P < 0.000$). Phenol oxidase activity also showed a strong correlation with pH across the seven sites ($R^2 = 0.610$, $P = 0.023$). However, neither temperature nor water level exhibited significant correlations with phenol oxidase activity across the locations. Phenol oxidase has been noted to play a key role in overall mineralization of organic matter in peatlands[1,9], and to be strongly controlled by pH[17,18]. Our results suggest that pH is the dominant controlling variable for phenol oxidase activity in peatlands at a global scale. This conflicts with data collected at smaller scales, for example in Canadian peatlands where phenol oxidase activity was mainly controlled by temperature and by a lesser extent with pH[19], or in nutrient-poor tundra soils where manipulating soil pH by liming did not affect phenol oxidase activities[20]. The mean temperatures are much lower in those previously studied sites where $Q_{10}$ values are also typically higher than warmer conditions[21]. In contrast, our results are based on far broader temperature ranges (Supplementary Table 1) and yet despite this pH emerged as the dominant driver.

Higher activities of phenol oxidase at higher pH can arise through either (a) activation of legacy enzymes already present in the soil matrix, or (b) increases in de novo synthesis of enzymes in the presence or absence of changes in microbial composition. Various studies have shown that phenol oxidase can be rapidly activated by pH changes and suggest that optimal pH of the enzyme is around neutral to alkaline pH[17]. Because most samples of this study were acidic, the activity of phenol oxidase may respond markedly to higher pH. In addition to the direct influence of pH on phenol oxidase activity, microbial composition appeared to be another factor that shapes the relationship between pH and phenol oxidase activity. In the three peatlands in Korea, pH exhibited a significant correlation with logarithmic gene copy number of the laccase, a proxy of phenol oxidase

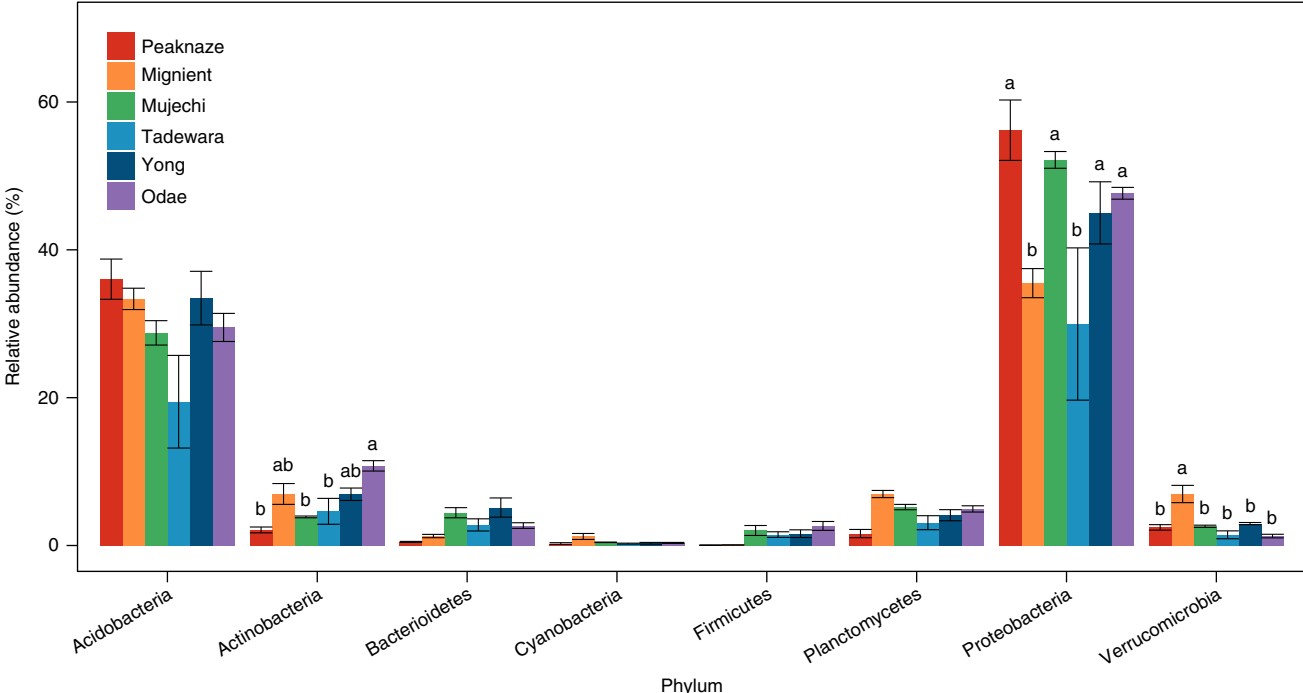

**Fig. 2** Relative abundances of bacterial community by phyla in six peatlands in Korea, Japan and UK. Relative abundances were determined by pyrosequencing using a Roche 454 Titanium platform and Ez-Taxon e-database. Error bars indicate the standard error of means

### Table 1 Microbial diversity and abundances in seven peatland sites

|  | Odae | Yong | Mujechi | Tadewara | Kalimantan | Migneint | Peaknaze |
|---|---|---|---|---|---|---|---|
| Bacterial diversity (T-RFLP) | 3.04 (0.03) | 2.93 (0.15) | 2.83 (0.05) | 3.11 (0.08) | 2.39 (0.09) | 3.19 (0.07) | 2.72 (0.05) |
| Fungal diversity (T-RFLP) | 1.45 (0.15) | 1.72 (0.13) | 2.43 (0.16) | 2.48 (0.22) | 2.05 (0.08) | 1.09 (0.07) | 2.51 (0.35) |
| Bacterial diversity (pyrosequencing) | 7.18 (0.04) | 6.94 (0.05) | 7.26 (0.03) | 4.75 (0.06) | ND | 6.55 (0.07) | 4.89 (0.27) |
| Bacterial abundance (copy # g soil$^{-1}$) | $2.50 \times 10^{10}$ ($4.02 \times 10^8$) | $1.36 \times 10^{10}$ ($3.03 \times 10^9$) | $8.29 \times 10^9$ ($1.54 \times 10^9$) | $1.43 \times 10^{10}$ ($4.14 \times 10^9$) | $9.73 \times 10^9$ ($1.42 \times 10^9$) | $3.31 \times 10^{10}$ ($0.05 \times 10^{10}$) | $5.67 \times 10^{10}$ ($1.42 \times 10^{10}$) |

Diversity was based on Shannon Index and abundance was measured by real-time qPCR

(Supplementary Fig. 1; $r = 0.577$, $P < 0.000$). The result of pyrosequencing showed a significant difference in the relative abundance of *Actinobacteria* in the six peatland sites in Korea, UK and Japan (Fig. 2). The difference was generally in accordance with a pH gradient; there was a higher relative abundance in *Actinobacteria* in higher pH conditions ($r = 0.738$, $P = 0.09$), suggesting that this phylum could be a driver of the greater phenol oxidase activity under high pH conditions. Indeed, it has been reported that filamentous *Actinobacteria* decompose lignocellulose by producing various enzymes including phenol oxidase[22] and they are also known to produce DOC in the form of soluble polyphenolics as end products of the decomposition[23]. It is also noteworthy that pH is the most significant edaphic factor for the Actinobacterial community in grassland soils[24]. A similar positive correlation between pH and the relative abundance of *Actinobacteria* has been found in 88 soils across North and South America[25].

Unlike the specific functional groups that are closely related to phenol oxidase such as abundances of *Actinobacteria* or laccase genes, no general patterns of bacterial diversity (determined by t-RFLP—terminal restriction fragment length polymorphism—or pyrosequencing) or abundance (real-time qPCR targeting 16S rRNA genes) across the seven sites were discernible (Table 1).

Likewise, fungal diversity estimated by t-RFLP patterns did not show any correlation with a pH gradient across the sites. However, we recognized that changes in fungal abundance could represent another potential driver[22], and so decided to monitor it in all future studies (Figs. 3, 4).

In each site, linear regression analysis revealed positive correlations between phenol oxidase activity and pore-water phenolic concentrations although they were marginally significant in Yong and Mujechi peatlands (Table 2), indicating that phenol oxidase activity indeed is associated with the release of phenolic materials from peat soils. The slopes of the regression lines (e.g., contribution of change in phenol oxidase to phenolic change) were higher in peatlands in Korea and Japan (0.231–2.691) than in the UK (0.091–0.100) or Indonesia (0.027), suggesting that the pH-mediated activation of phenol oxidase would be far more prominent in East Asian peatlands. Unlike decreases in acid deposition and the consequent recovery from acidification reported in Europe and North America[26], East Asian countries are still exposed to high acid deposition that mostly originates from China[27]. Clearly, in future, East Asian peatlands are far more at risk of rapid increases in DOC export than other regions, should (as occurred in Europe and North America in the closing decades of the last century) any

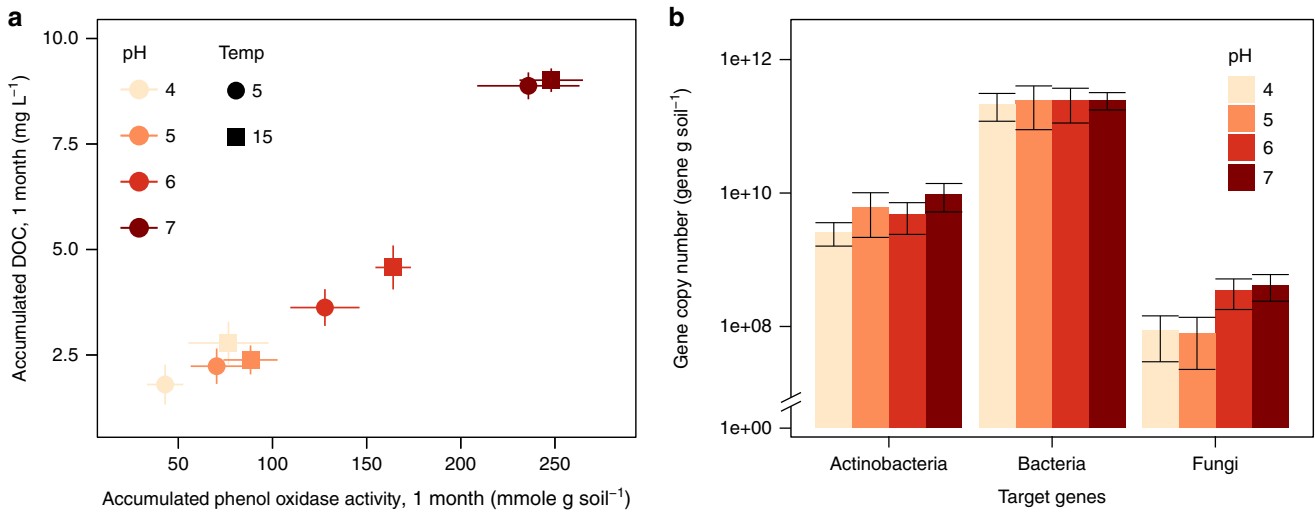

**Fig. 3** pH effects on phenol oxidase and gene copy numbers in the field. Relationship between pH and phenol oxidase activities in Migneint (**a**) and Peaknaze (**c**), and the gene copy numbers of bacteria, *Archaea* and fungi in different treatments in Migneint (**b**) and Peaknaze (**d**)

**Fig. 4** pH effects on phenol oxidase and gene copy numbers in the lab incubation. Relationship between accumulated DOC and phenol oxidase activities at different pH and temperatures (**a**) and the abundances of microorganisms (**b**) in peat soils incubated over a month-period incubation

| | Odae | Yong | Mujechi | Tadewara | Kalimantan | Migneint | Peaknaze |
|---|---|---|---|---|---|---|---|
| **Table 2 Results of linear regressions between phenol oxidase activity in peat matrix and the concentrations of phenolics in water in each wetland** | | | | | | | |
| $R^2$ | 0.206 | 0.129 | 0.149 | 0.339 | 0.142 | 0.096 | 0.064 |
| Slope | 2.691 | 1.887 | 0.233 | 0.231 | 0.027 | 0.091 | 0.100 |
| $P$ value | 0.023 | 0.092 | 0.084 | 0.023 | 0.040 | 0.007 | 0.028 |
| $N$ | 24 | 22 | 20 | 15 | 30 | 76 | 76 |

regulations and programs for the reduction in acid deposition be introduced in the region. However, under the current "business as usual" scenario, our results indicate that globally prevalent DOC trends could soon reverse due to biological responses to long range transport of atmospheric acidification from the unprecedented rate of industrialisation in China.

It should be noted that phenol oxidase activities were the only variable that exhibited a significant positive correlation with phenolic content in pore-water across all sites, while temperature, water level, inorganic nitrogen content and other factors, which are known to be critical for decomposition rates, failed to reveal correlations with phenolics. This result indicates a strong linkage from pH changes through phenol oxidase activity to phenolic release across a wide range of peatlands.

**pH manipulation experiments**. In addition to the above survey, further evidence was sought for pH-mediated effects in field and laboratory manipulations. A field pH manipulation experiment undertaken in UK for 4 years revealed higher phenol oxidase activity to be associated with higher pH (Fig. 3c; $R^2 = 0.327$, $P < 0.01$), although three outliers in one of the treatments prevented the observation of statistical significance in one site (Fig. 3a). This was accompanied by changes in the abundance of fungi (Fig. 3b, d). Additionally, a significant correlation was found between phenol oxidase activity and the relative abundance of *Actinobacteria* in Migneint samples ($r = 0.668$, $P < 0.05$) and a marginally significant correlation between phenol oxidase activity and fungal abundance in Peaknaze ($r = 0.514$, $P = 0.087$), indicating that pH increases would directly enhance phenol oxidase activities as well as indirectly by modifying microbial abundances, particularly *Actinobacteria* and fungi.

The field observations were supported by our laboratory manipulation experiment where incubation under elevated pH resulted in clear increases in the activities of phenol oxidase, and the concentrations of end product phenolics and DOC (Fig. 4a). Temperature appeared to be a secondary factor in the control of phenol oxidase activity: At each pH, samples incubated at 15 °C exhibited the higher activity of phenol oxidase and DOC concentrations than those incubated at 5 °C. Responses of hydrolases or $CO_2$ emissions to pH were much smaller or not significant, indicating that phenol oxidase activities and DOC leaching were most strongly and most rapidly affected by increases in pH while their effects on heterotrophic respiration were minimal (Supplementary Fig. 2), possibly due to the inhibitory effects of the released phenolics[9]. Concomitantly, the gene copy numbers of *Actinobacteria* and fungi were higher under higher pH although the differences were not statistically significant due to huge variations among samples of the same treatments (Fig. 4b). However, a simple correlation analysis with the whole data set showed significant correlations between phenol oxidase activity and the abundances of *Actinobacteria* ($r = 0.584$, $P < 0.05$) and fungi ($r = 0.595$, $P < 0.05$), but not with bacteria ($r = 0.111$, $P = 0.731$).

The tendency of greater abundances in *Actinobacteria* and fungi, and significant positive correlation with phenol oxidase

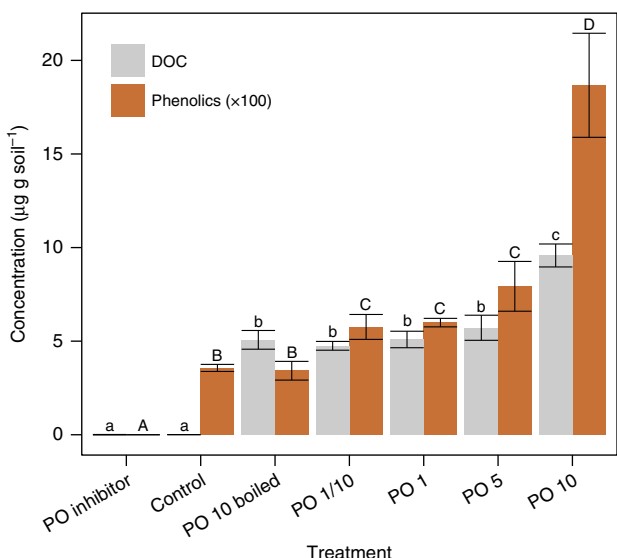

**Fig. 5** Effects of phenol oxidase additions or inhibition on DOC and phenolic concentrations in peat slurry. Different letters represent significant differences in DOC (lower case) or phenolics (upper case) among different treatments

suggest that those microorganisms are partially responsible for the greater activities of phenol oxidase under higher pH conditions. Overall, our results suggest that pH changes regulate legacy phenol oxidase in the soil matrix instantly, and can be reinforced by longer term increases in *Actinobacteria* and fungi abundances as the latter can further enhance the production of phenol oxidase.

**Enzyme manipulation experiment**. Although our field observations and pH manipulation experiments suggest that pH increases activate phenol oxidase, and that this is subsequently associated with DOC and phenolics release, they do not provide proof of causation. Our final enzyme manipulation experiment allowed us to confirm that the activation of phenol oxidase could enhance phenolics and DOC leaching from the peat matrix. Phenol oxidase addition increased the concentrations of DOC (at 10-unit enzyme addition) and phenolic (from 0.1 unit enzyme addition) (Fig. 5). In contrast, addition of an inhibitor substantially decreased DOC and phenolics mobilisation, supporting that the enzyme reaction is directly associated with the production of DOC and phenolics in the system. Contributions from direct degradation of phenol oxidase added were minimal as the control (pre-boiled) sample did not show a significant increase in phenolics although it affected DOC concentration (Fig. 5).

This study, based on a field-based global survey as well as manipulations in both field and laboratory settings, clearly demonstrates that pH increases in peatlands can accelerate DOC leaching from the global peatland carbon store by a direct edaphic stimulation of phenol oxidase, with the potential

for further sustained contributions from more abundant *Actinobacteria* and fungi in peat soils in the longer-term. The results further suggest that the possible transfer of carbon storage from terrestrial ecosystems to aquatic systems by acid deposition-mediated changes in phenol oxidase activity and fungal/Actinobacterial abundances should be considered as a potential mechanism for inclusion in global carbon modelling, alongside processes accelerated by global warming and precipitation change.

## Methods

**Field sampling and analysis**. Three mountain peatlands were selected in Korea, which have a similar vegetation structure (*Sphagnum*-dominated) but differ in the amount of acid deposition and pore-water pH. They are located at high altitude, are deprived of human intervention, and are isolated from watershed input with sole influence from atmospheric deposition. In Korea, samples were collected in summers and winters of 2009, 2010 and 2013 to cover seasonal variations. In each wetland, we collected samples from at least three different locations (the center, marginal area and the very edge of peatland) to incorporate spatial heterogeneity of edaphic and hydrological conditions. Pore-water samples were collected at 10 cm depth from the surface. A total of 20–22 field surveys were conducted in each site. In addition, 5-year field monitoring study was conducted at two sites in the UK. One site was in Migneint, Wales, and the other one was in Peaknaze, Peak District. Details of the sites are available in a previous report[2]. Three replicate samples were collected from August 2007 to June 2012 monthly or biweekly, resulting in 76 sampling occasions. A similar field survey was conducted in Tadewara mire, Oita, Japan, of which the main vegetation was *Sphagnum palustre, S. fimbriatum* with *Phragmites australis*. The sampling was conducted in May 2013. It was temporally limited compared with the other sites, but samples were collected from six different locations from the centre of *Sphagnum*-dominated site to upland fringe to cover the spatial heterogeneity of the site. For tropical peatland, peat cores were collected in peatland near Hampangen in Kalimantan, Indonesia, where a total of 30 samples were collected from three locations with different drainage histories to ensure robust spatial coverage. The main vegetation was *Combretocarpus rotundatus*, *Cratoxylum glaucum*, *Palaquium leiocarpum* and *Syzygium creaghii* depending on water level which fluctuated between 0 and 20 cm below the surface. Details of each site are presented in Supplementary Table 1.

For soil sampling, the surface living part of peat was carefully removed with a knife and a shovel, and peat samples beneath it (0–10 cm depth from the surface excluding the living part of *Sphagnum*) were collected. All samples were delivered to the lab on ice, and subsamples were maintained at 4 °C for chemical analysis and enzyme activity measurement, or deep-frozen to −30 °C for molecular analysis of microorganisms.

We measured pH, temperature, and conductivity in situ, while water samples were collected at a depth of 10 cm from the peat surface for the measurements of concentration of ions, DOC, and phenolic materials in the lab. Both cations and anions were measured by ion chromatography (Dionex, Sunnyvale, USA), DOC with a total organic carbon analyser (Shimadzu, Kyoto, Japan) and phenolics by Folin reagent with filtered water samples.

Phenol oxidase activity was measured using of L-DOPA (L-3,4,-dihydroxyphenylalanine) as a model substrate[17]. Hydrolases (ß-glucosidase, N-acetylglucosaminidase, phosphatase and arylsulfatase) were determined using methylumbelliferyl compounds as model substrates[28].

The DNA was extracted from peat soils with a Power soil DNA Kit (MoBio, Carlsbad, USA) according to the manufacturer's instructions. Nucleic acid quality and quantity were assessed by using a NanoDrop ND-1000 Spectrophotometer (NanoDrop Technologies Inc., Wilmington, USA). We performed a pyrosequencing analysis to expand information of the bacterial community in the soil samples. Two template DNA from two composite soil DNA samples were used for pyrosequencing analysis. All analyses including profiling sequence data according to Ez-Taxon e-database (http://www.ez-taxon-e.org) were performed at Chunlin Inc. (Chunlab Inc., Korea, http://www.chunlab.com) which provides pyrosequencing services with a Roche 454 GS FLX Titanium platform. To estimate gene copy numbers of bacteria, fungi, *Archaea*, Actinobacteria and laccase, we performed qPCR using CFX96 (Bio-Rad, Hercules, CA, USA) and SYBR Green as a detection system (Toyobo, Japan). Each 20-μl reaction reagent contained the specific primers. Two independent real-time PCR assays were performed on each soil DNA extract.

For t-RFLP (terminal restriction fragment length polymorphism) analysis, the pooled PCR products were purified and added to a reaction mixture containing *Hha*I. The terminal fragment size analysis was performed using an ABI 3730 DNA Analyser (Applied Biosystems, Foster City, USA) in conjunction with GeneScan software (Applied Biosystems, Foster City, USA). Terminal restriction fragments (TRFs) were quantified via peak area integration using a minimum peak height threshold of 50 relative fluorescent units. The dominant TRFs in each microbial TRFLP profile were inferred by in silico digestion of sequences of each gene present in the database.

**Field pH manipulation experiment**. The experiment of pH manipulation in the field conditions was undertaken on peat soils at both the Peak district in Northern England (higher historic acid deposition) and the Migneint in North Wales (lower historic acid deposition). At both sites, twelve 3 × 3 m plots were assigned to control, acid and alkaline treatments in each peatland site and added with rain water, rain water plus $H_2SO_4$, and rain water plus NaOH and $MgCl_2$, respectively[2]. The pH manipulation treatments lasted from 2007 to 2012, and peat soil samples were collected in July and September 2012. Phenol oxidase and microbial properties were measured previously described.

**Laboratory pH and temperature manipulation experiment**. Peat samples were collected from three locations in Yong peatland (which is the mid-pH site of the three sites in Korea). Peat samples were gently homogenized and 250 g subsamples were placed into glass jars (600 mL). Peat slurry pH was modified to 4, 5, 6 and 7 by adding drops of 1 M HCl or NaOH with continuous stirring. Four jars at each acidity were then incubated at 5 °C while the other four jars were incubated at 15 °C. Filtered field water was added regularly to compensate water loss by evaporation.

Five grams of soil slurries were sampled after 0, 2, 7, 14 and 28 days. At the same time, gas samples were collected by closing the incubation jar and taking gas samples five times from the headspace over 12-min intervals. Soil analysis was conducted with the same methods described above. The concentrations of $CO_2$, $CH_4$ and $N_2O$ were analysed by Gas Chromatography equipped with FID (flame ionization detector) and ECD (electron capture detector). All microbial analyses were conducted using aforementioned methods.

**Laboratory enzyme manipulation experiment**. Peat slurry was prepared by mixing peat and water collected from the sites (80% of water content). Purified phenol oxidase (laccase purified from *Trametes versicolor*; Sigma-Aldrich, St. Louis, USA) was dissolved in buffer solution. One unit of the enzyme corresponds to the amount of enzyme which converts 1 μmol of catechol per minute at pH 6.0 and 25 °C.

Twenty-five grams of peat slurry were combined with 5 mL of enzyme solutions with 0.1, 1, 5, or 10-unit enzyme. In addition to enzyme treatments, (1) a water-only addition was included as a control, (2) a set with pre-boiled enzyme solution (10 units) was prepared to consider the contribution of DOC or phenolics from the degradation of enzyme protein itself, and (3) sodium diethyldithiocarbamate was added, which is known to completely inhibit phenol oxidase activity[29]. Three replicate samples were prepared and incubated at 25 °C for 18 h. Samples were then filtered and the concentrations of anions, cations, DOC and phenolics were measured as described above.

A correlation analysis, a linear regression, or a one-way ANOVA test was applied to corresponding data sets (SPSS, Version 21).

## Data availability

The data sets and Supplementary Information of the current study are available from the authors upon request. The dataset of 16S rDNA gene amplicon sequences reported in this study has been deposited at GenBank (https://www.ncbi.nlm.nih.gov/genbank/) and their accession numbers are available in Supplementary Table 2.

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

## Acknowledgements

We are grateful to NRF (20110030040, 2016R1D1A1A02937049, 2016M1A5A1901795) and Korea Forest Service (2017096A001719BB01) for financial support. We also thank the Natural Environment Research Council, UK, for support. In particular, we thank the staff of the Centre for Ecology and Hydrology (including A. Burden, N. Ostle & C. Evans) in relation to a NERC grant involving CF & TJ (NE/E011748/1; 2007-2010), which established the sites from which the UK samples were subsequently collected.

## Author contributions

H.K. conceived and co-designed the study, collected field data and led the writing of the paper. C.F. co-conceived and co-designed the study and contributed to data collection and writing. M.J.K contributed to the writing, field survey, data collection and analysis. S.K. contributed to the data collection and analysis, field survey and writing. S.L. contributed to the data collection and analysis and writing. T.G.J. contributed to data collection, field survey and writing. A.C.J. contributed to data collection and writing. A.H. contributed to field data collection, analysis and writing.

## Additional information

**Competing interests:** The authors declare no competing interests.

