## [Peer Review File · Nature Communications]

REVIEWERS' COMMENTS:

Reviewer #1 (Remarks to the Author):

The study is thorough and persuasive: it effectively integrates much of the confusing and seemingly contradictory literature on edaphic and enzymatic controls on peatland carbon dynamics by connecting environmental responses (i.e. drying, temperature, pH) to a phenol oxidase "latch" as proposed by Freeman et al. 2001. The paper is clearly written. The methods and data analyses are sound. The paper is a significant contribution to the larger topic of soil carbon storage. It is acceptable for publication as written.

Reviewer #2 (Remarks to the Author):

This manuscript uses a variety of approaches to examine the pH control of DOC concentrations via phenol oxidase activity and microbial activity. These approaches include long-term observations in 7 peatlands in Korea, the UK, Japan, and Indonesia, pH manipulation experiments in 2 peatlands, a laboratory experiment where pH and temperature were manipulated, and a laboratory experiment where phenol oxidase activity was manipulated. This very thorough approach shows that pH directly controls phenol oxidase activity, and thus DOC concentrations, but that also there is a pH control of Actinobacteria and fungi abundance that provides a biological control of phenol oxidase activity.

This is an excellent study in terms of its thoroughness and multiple approaches, and it provides strong support that amelioration of atmospheric acid deposition has led to the observed increases in DOC concentrations in surface waters due to the increase in pH in peatlands. It also provides multiple mechanisms for this effect, and according to the manuscript (and I have reason not to believe the assertion), this is the first study to show a biological mechanism for this phenomena.

I have only relatively minor criticisms of this manuscript and consider it very worthy of publication in Nature Communications.

Specific Comments

The preference of much of the scientific community is to use the term dissolved organic matter rather than dissolved organic carbon because the former is more generic and DOM contains more than C. Concentration of DOM is measured as DOC.

Line 35. Alkaline is an adjective, so you should use alkali here.

Lines 73-76. It's not completely clear to me but two different regressions are referred to and one figure. I assume that the second one is where phenol oxidase activity is not logged. This is confusing and the authors should pick whichever is the best relationship based upon linearity and the r^2 .

Lines 140-142. This statement ignores the lack of pH effect on phenol oxidase activity in panel A. A more guarded interpretation is appropriate here.

Lines 178-180. I may be missing something here, but the boiled samples do appear to have an increase in DOC over the control sample in Fig. 5.

Lines 195-196. How do the authors know that the peatlands are ombrotrophic? The low pH certainly suggests this, but this is a very definitive statement. Have they done hydrology studies? If not, then this is another place where they should be a bit more circumspect in their language.

Reviewer #1 Stated that;

The study is thorough and persuasive.....The paper is a significant contribution to the larger topic of soil carbon storage. It is acceptable for publication as written.

→ **We are grateful to the reviewer for the positive comments.**

Reviewer #2 Stated that;

This is an excellent study in terms of its thoroughness and multiple approaches..... according to the manuscript (and I have reason not to believe the assertion), this is the first study to show a biological mechanism for this phenomena.

→ **We are grateful to the reviewer for the positive comments.**

The preference of much of the scientific community is to use the term dissolved organic matter rather than dissolved organic carbon because it the former is more generic and DOM contains more than C. Concentration of DOM is measured as DOC.

→ **We agree to the point and added following sentence.**

“DOC is a part of dissolved organic matter (DOM) which also includes other materials such as dissolved organic nitrogen (DON) and phosphorus (DOP). However, DOC and DOM are often interchangeably used and measured by the same method.”

Line 35. Alkaline is an adjective, so you should use alkalai here.

→ **Corrected accordingly.**

Lines 73-76. It's not completely clear to me but two different regressions are referred to and one figure. I assume that the second one is where phenol oxidase activity is not logged. This is confusing and the authors should pick whichever is the best relationship based upon linearity and the r2.

→ **The reviewer made a valid point. To clarify it, we deleted 'Fig. 1' from the second sentence which doesn't refer to Figure 1.**

Lines 140-142. This statement ignores the lack of pH effect on phenol oxidase activity in panel A. A more guarded interpretation is appropriate here.

→ **We agree to the point and added following sentence.**

“A field pH manipulation experiment undertaken in the UK for 4 years revealed higher phenol oxidase activity to be associated with higher pH (Fig. 3-c; $R^2 = 0.327$, $P < 0.01$), although 3 outliers in one of the treatments prevented the observation of statistical significance in one site (Fig 3-a).”

Lines 178-180. I may be missing something here, but the boiled samples do appear to have an increase in DOC over the control sample in Fig. 5.

→ **The reviewer made a valid point and modified it as follows.**

“Contributions from direct degradation of phenol oxidase added were minimal as the control (pre-boiled) sample did not show a significant increase in phenolics although it affected DOC concentration (Fig. 5).”

Lines 195-196. How do the authors know that the peatlands are ombrotrophic? The low pH certainly suggests this, but this is a very definitive statement. Have they done hydrology studies? If not, then this is another place where they should be a bit more circumspect in their language.

→ **We agree to the point and deleted ‘ombrotrophic’ from the sentences as we do not have full hydrology data for the sites in Indonesia and Japan.**